# Quality by Design Guided Development of Polymeric Nanospheres of Terbinafine Hydrochloride for Topical Treatment of Onychomycosis Using a Nano-Gel Formulation

**DOI:** 10.3390/pharmaceutics14102170

**Published:** 2022-10-12

**Authors:** Vinam Puri, Anna Froelich, Parinbhai Shah, Shernelle Pringle, Kevin Chen, Bozena Michniak-Kohn

**Affiliations:** 1Department of Pharmaceutics, Ernest Mario School of Pharmacy, Rutgers, The State University of New Jersey, Piscataway, NJ 08855, USA; 2Center for Dermal Research, Life Science Building, Rutgers, The State University of New Jersey, Piscataway, NJ 08854, USA; 3Chair and Department of Pharmaceutical Technology, Poznan University of Medical Sciences, Grunwaldzka 6, 60-780 Poznan, Poland; 4Department of Biomedical Sciences, School of Graduate Studies, Rutgers, The State University of New Jersey, Piscataway, NJ 08855, USA

**Keywords:** antifungal nanospheres, ungual drug delivery, quality by design, design of experiments, onychomycosis, topical formulation, terbinafine

## Abstract

Superficial fungal diseases of the skin and nails are an increasingly common occurrence globally, requiring effective topical treatment to avoid systemic adverse effects. Polymeric nanoparticles have demonstrated sustained and effective drug delivery in a variety of topical formulations. The aim of this project was to develop polymeric antifungal nanospheres containing terbinafine hydrochloride (TBH) to be loaded into a hydrogel formulation for topical nail drug delivery. A quality by design (QbD) approach was used to achieve optimized particles with the desired quality target product profile (QTPP). Polyvinyl alcohol (PVA) at 2% *w*/*v* and a drug to polymer ratio of 1:4, together with a robust set of processes and material attributes, resulted in nanoparticles of 108.7 nm with a polydispersity index (PDI) of 0.63, 57.43% recovery, and other desirable characteristics such as zeta potential (ZP), particle shape, aggregation, etc. The nanospheres were incorporated into a carbomer-based gel, and the delivery of TBH through this formulation was evaluated by means of in vitro drug release testing (IVRT) and ex vivo nail permeation study. The gel containing the TBH nanospheres demonstrated a slower and controlled drug release profile compared with the control gel, in addition to a more efficient delivery into the nail. These antifungal nanospheres can be utilized for topical therapy of a multitude of superficial fungal infections.

## 1. Introduction

Fungal infections are the most common type of skin disease worldwide, superficial infections being in the top 10 most prevalent [1,2]. Onychomycosis, a type of superficial fungal infection, is the most common disorder of the nail [3,4]. It is also said to be the most difficult superficial fungal infection to cure, and accounts for about 50% of all nail diseases [5]. In immunocompromised patients, fungal infections have become one of the major causes of morbidity and mortality [6]. Often found in the outer layers of the skin, nails, hair, and mucous membranes, such infections present an enormous challenge to healthcare professionals [7]. Skin infections may cause inflammation, pain, itchiness, redness, cracking and softening of the skin [1]; while the nail disease presents with discoloration, thickness and brittleness of the nail [8]. Presence of one type of fungal infection often leads to other fungal infections and is common in skin and nail diseases [9]. Although rarely life threatening, these diseases have been shown to have psychological, social and occupational impacts, and result a reduction in life quality for patients [7,10].

Treatment of superficial fungal diseases is challenging due to factors such as uncertainty of treatment duration, high relapse rates, and adverse effects of systemic treatments [5,11]. Systemic antifungals, on top of carrying risk of toxicity, also possess the problem of drug–drug interactions [12]. Topical therapy is more effective in skin applications compared with nail disease, for which combination treatment using simultaneous or subsequent administration of oral and topical agents is often used [11,13]. The biggest challenge with topical therapy is its limited effectiveness, possibly due to failure to achieve effective concentrations of the drug in the infected tissue, as well as maintaining drug concentrations locally [13]. Combination therapy with oral and topical antifungals has shown more success against disease compared with monotherapy [14]. Although combination therapy aims at minimizing the disadvantages of both types of treatments and providing their advantages, such as antimycotic synergy, enhanced tolerability and safety [15], an effective topical monotherapy at the site of infection would be preferred and more accepted due to ease of administration and lack of adverse effects. Drug-resistance in fungal infections is another growing challenge, although not highly significant in superficial infections, but concerning nevertheless [16].

The ineffectiveness of topical nail formulations can be attributed to limited or less sustained delivery of drugs through the highly keratinized nail plate [17]. The slow growth rate and low permeability of nails warrant longer and more effective treatment options [18]. Different methods of permeation enhancement have been employed to achieve topical drug delivery including physical methods such as nail avulsion, surface etching, microporation, laser, photodynamic therapy, UV irradiation and iontophoresis; and chemical penetration enhancers such as keratolytic chemicals, enzymes, surfactants, proteins, solvents, etc. [19]. Hydration of the nail plate has been shown to have a significant impact on its permeability [20]. Recent formulation approaches to improve topical permeation of antifungal drugs include transferosomes, micro/nano-emulsions, and liposomes delivered via carrier systems such as lacquers, gels, patches and films [21].

Terbinafine is a well-known lipophilic broad spectrum allylamine antifungal which is also marketed in oral and topical formulations [1,22,23] and has been found to be very potent not only against *Trichophyton rubrum*, but also a broad range of fungi, yeast, molds and dermatophytes [24]. Its mode of action is by inhibiting formation of squalene epoxidase enzyme which enables the ergosterol pathway leading to fungistatic and fungicidal effects [25]. Terbinafine has also shown to be more efficient than some antifungals in clinical studies [26]. Formulations with terbinafine range from simple creams [16] to microemulsion-based gels [27], liposomal film formulations [28], bilayered lacquers [29], and liposomes- and ethosomes-loaded gels [30]. Small particles that can stay in the *stratum corneum* and skin pores and folds over a prolonged period [31] can permeate better, and hence, can deliver drugs more effectively. Such particles have been used for pharmaceutical as well as cosmetic delivery of actives in a controlled manner [6,12,32,33].

Quality by design (QbD) is a risk-assessment-based systemic approach for complex formulations involving several parameters, that assists in understanding the sources of variability during product formulation [34,35]. Topical formulations have been optimized and efficiently produced utilizing this approach to achieve the quality target product profile (QTPP) [33,36,37]. The FDA has been promoting the use of QbD for pharmaceutical product development, and ICH Q8, Q9, and Q10 guidelines have been published by the International Conference on Harmonization to assist with the implementation of this approach to achieve quality pharmaceutical products [38,39,40]. Statistical design of experiments (DoE) can be used to study quality characteristic responses to variations in parameters that play a role in the formation of the product [35]. The understanding from DoE is based on mathematical relationships between formulation inputs and outputs [35,41].

The current work involves the development of terbinafine hydrochloride (TBH)-loaded polymeric nanospheres, optimized using a quality by design (QbD) guided approach. The desired quality target product profile (QTPP) of the nanospheres was achieved by optimizing the process and material variables that impacted the critical quality attributes (CQAs) of the particles. Optimized nanospheres were incorporated into a carbomer-based hydrogel as a delivery vehicle to achieve hydration of the nail plate and controlled delivery of TBH from the nanospheres. It was hypothesized that the positively charged nanospheres would lodge into the structural deformities of hydrated onychomycotic nails, owing to their small size and potential bioadhesivity on the negatively charged nail plate, favoring nail diffusivity [42,43]. The antifungal nanosphere-loaded gel drug delivery system was analyzed for in vitro drug release and ex vivo nail permeation in comparison with a control TBH gel, to investigate its capability of achieving the desired controlled drug delivery. The obtained results support the potential of this ungual drug delivery system to provide an effective treatment approach for the management of onychomycosis. Such a drug delivery system could be optimized to help other challenging topical therapies and minimize the undesirable effects of systemic therapy.

## 2. Materials and Methods

### 2.1. Materials

Terbinafine hydrochloride (95%; TBH) was purchased from AstaTech, Inc. (Bristol, PA, USA). Ethyl cellulose (EC) and Eudragit RSPO (E-RSPO) were kind gifts from BASF (Tarrytown, NJ, USA) and Evonik Corporation (Piscataway, NJ, USA), respectively. Dichloromethane (DCM), poly(vinyl alcohol) (m.w. 13,000–23,000, 87–89% hydrolyzed; PVA), ethyl alcohol, triethanolamine, thioglycolic acid, polyethylene glycol 400, polysorbate 80 (Tween 80), formic acid, HPLC grade acetonitrile, and water, were purchased from Sigma-Aldrich, Inc. (St. Louis, MO, USA) and used as received. Dulbecco’s phosphate buffered saline (DPBS) was purchased from Thermo Fisher Scientific (Waltham, MA, USA). Carbopol^®^ Ultrez-10 was a gift from Lubrizol Corporation (Wickliffe, OH, USA). Phosphate buffer saline (PBS) tablets were purchased from Tocris, Bio-Techne Corporation (Minneapolis, MN, USA).

### 2.2. Preparation of TBH Nanospheres

Particles were prepared with the modified method described by Božič et al. [44]. Briefly, the polymer and the drug were dissolved in DCM. Next, the organic phase (internal phase) was added to PVA solution (external phase) and mixed with a magnetic stirrer at rate of 450 rpm in a closed vial for 10 min. The resulting oil/water emulsion was ultrasonicated with a Branson Digital Sonifier SFX 150 (Emerson Electric Co., St. Louis, MO, USA) equipped with a microtip probe, in pulse mode with 0.7 s on and 0.3 s off, at 70% output energy. The emulsion was then left uncapped, stirring at 450 rpm at room temperature under a chemical fume hood overnight, for residual solvent evaporation. The scheme depicting basic steps of the particle formation is presented in Figure 1. Following solvent evaporation, the suspension was transferred to 13.5 mL Quick-Seal^®^ polypropylene tubes (Beckman Coulter, Brea, CA, USA). The tubes were filled with deionized water and sealed. The samples were centrifuged at 20,000 rpm for 15 min at 18 °C with an Optima™ L-90K Ultracentrifuge (Beckman Coulter) equipped with a 70.1 Ti rotor. Next, the supernatant was discarded, and the pellet was collected, redispersed in deionized water, and centrifuged again with the same procedure. After the removal of the supernatant, the pellet was transferred to a vial, redispersed in 5 mL of deionized water, and shaken overnight to obtain an aqueous suspension of the drug-loaded nanospheres.

### 2.3. Quality by Design (QbD) Steps

#### 2.3.1. Quality Target Product Profile (QTPP)

A summary of the desired quality characteristics of the nanospheres was listed, in order to identify the quality attributes that are critical for the product. To establish the QTPP and CQAs, information from scientific literature, prior knowledge about the scientific, regulatory and practical considerations for formulation, and preliminary experimentation, were used. The QTPP and quality attributes were listed and CQAs for the nanospheres were identified and summarized in tabular form, with justifications for each quality attribute.

#### 2.3.2. Risk Assessment

A cause-and-effect diagram (Ishikawa diagram) was employed to ensure the capture of all possible factors, such as materials and processes, that could affect product quality. The initial risk assessment involves estimation of risk from the parameters identified in the Ishikawa diagram on the CQAs. A risk assessment table was created to assess the level of impact that all identified factors could have on the quality of the product.

#### 2.3.3. Design of Experiments (DoE)

The factors that were identified to carry higher levels of risk on product quality were screened through experiments conducted using DoE. Preliminary screening experiments were conducted with the goal of understanding the effect of mixing time and sonication time as process parameters; and to select a polymer nanosphere formation. JMP^®^ (SAS Institute Inc., Cary, NC, USA) was employed for generation of randomized experimental design tables. Two 2^4^ full factorial designs were employed for each of the two polymers utilized—EC and E-RSPO. The material variables were drug to polymer ratio and PVA concentration; and the process variables were mixing time after primary emulsion formation and total sonication time during ultrasonication. Internal phase volume was kept constant at 2 mL and external phase volume was 10 mL. Table 1 shows the low and high levels at which the variables were tested for the full factorial design of experiments. The product quality attributes (responses) characterized during these preliminary experiments to study the impacts of the independent variables were average particle size (Z-ave) and polydispersity index (PDI). These preliminary trials were used to guide further experimental designs.

Based on the results from the preliminary experimental design, a secondary fractional factorial design (Table 2) guided by response surface methodology (RSM) was used to investigate the effect of changing EPV to IPV ratios while still using Z-ave and PDI as responses for this set of experiments. This was used to minimize the experimental trials while evaluating main and interaction variable effects at low, medium and high levels.

The preliminary and secondary experimental trials provided the required information to design the final DoE that included PVA concentration and drug to polymer ratio as the independent variables at three levels each. The PVA concentration was varied at levels of 0.25, 1 and 2% *w*/*v*, and the drug to polymer ratio was varied at levels of 1:4, 1:1 and 4:1, making a 3^2^ factorial design. A power analysis was performed to obtain the minimum number of replicates and center points, hence experimental trials needed to obtain a statistically confident outcome. For this analysis, a 20-run design was compared against 18, 22, 24, and 27-run designs that were obtained by changing the number of replicates and center points. This resulted in a 3^2^ factorial design matrix with 20 experimental runs containing duplicate trials and one center point. Table 3 summarizes the variable levels, responses and experimental design specifications used in the final design. The responses included in this optimization included recovery (%) along with Z-ave (nm) and PDI.

The parameters that were maintained at constant levels along with their values are listed in Table 4. These factors were the weight of the solids used for the nanosphere synthesis, the volumes of IPV and EPV, mixing time, mixing speed, sonication time, and sonication power; and their values were selected based on the observations from preliminary experiments as factors that either did not impact or were necessary for the product quality. The design analysis was performed using a linear regression model as described in Equation (1).
(1)$$Y=\beta_{0}+\beta_{1}X_{1}+\beta_{2}X_{2}+\beta_{12}X_{1}X_{2}+\varepsilon$$
where *Y* represents the response, which denotes each of the quality attributes of the product; *X*_1_ and *X*_2_ represent the main effects of the factors being varied; *X*_1_*X*_2_ represents the interaction effect of the independent variables; *β*_0_ denotes the arithmetic mean of quantitative outcomes from all the experimental runs; and *β*_1_, *β*_2_ are estimated coefficients from observed values of Y for both the factors. Based on Table 3 above, linear regression analysis was performed for each of the three responses—*Y*_1_ (Z-ave), *Y*_2_ (PDI), and *Y*_3_ (recovery); while *X*_1_ and *X*_2_ were the PVA concentration and drug to polymer ratio, respectively.

Finally, the optimum formulation prediction from the regression analysis of the model against the data obtained from the trials was used to validate the model and verify the accuracy of the predictions. The average responses from triplicates of this optimum combination were compared against the model-predicted responses under maximum desirability.

### 2.4. Particle Size Analysis

The average size and polydispersity of the particles in suspension were measured by dynamic light scattering (DLS). Approximately 0.1 mL of the particle suspension was diluted with 0.5 mL of deionized water and placed in polystyrene cuvettes. The measurements were performed at ambient temperature with Zetasizer Nano S (Malvern, UK) utilizing a 632.8 nm helium–neon laser. The scattered light was analyzed at an angle of 173° with the use of non-invasive backscatter mode. All samples were equilibrated for 120 s and analyzed in triplicate, with each repetition consisting of 15 scans. The obtained correlogram, plotted as a correlation function vs. time, was processed with Malvern Zetasizer software v. 7.12 to calculate Z-Ave as intensity weighed mean hydrodynamic size of the investigated particles, as well as polydispersity index (PDI).

### 2.5. Estimation of Recovery

Solid residue obtained from the drying of 1 mL of well-dispersed nanosphere suspension was weighed and dissolved in 1 mL of the high-performance liquid chromatography (HPLC) mobile phase consisting of 0.1% formic acid water solution and acetonitrile with 0.1% formic acid (65:35, *v*/*v*). The solutions were filtered with 0.45 μm PTFE filters and the drug content was analyzed with the validated HPLC method. TBH recovery in the nanospheres was calculated according to Equation (2).
(2)$$Recovery=\frac{Actual\;drug\;content\;in\;nanoparticles}{Theoretical\;drug\;content\;in\;nanoparticles}\times 100\%$$

### 2.6. Zeta Potential (ZP) Analysis

The measurements for ZP were performed using a Zetasizer Nano ZS90 (Malvern, UK) equipped with a 633 nm helium–neon laser and utilizing laser Doppler micro-electrophoresis technique. The samples were placed in a folded capillary cell (DTS1070) and the air bubbles were removed. The samples were equilibrated for 120 s prior to measurement, which was conducted at 25.0 ± 0.1 °C. Each measurement consisting of 12 runs was performed in triplicate and the results were presented as mean values.

### 2.7. Differential Scanning Calorimetry (DSC)

Physical mixtures containing TBH and E-RSPO at 4:1, 1:1 and 1:4 weight ratios, as well as solid residues obtained from drying 1 mL of optimized nanosphere suspensions, were investigated in DSC studies. The samples (1–3 mg) were placed in aluminum 40 μL crucibles with pierced lids. The experiments were performed with the DSC 823^e^ (Mettler Toledo, Columbus, OH, USA) differential scanning calorimeter equipped with a TSO801RO sample robot and Julabo FT 100 intracooler. During the experiment, the temperature was increased linearly from 25 to 230 °C at a rate of 10 °C min^−1^ under a constant flow of nitrogen (30.0 mL min^−1^). An empty crucible was used as a reference. The obtained thermograms were analyzed with STARe Software version 9.10 (Mettler Toledo, Columbus, OH).

### 2.8. Transmission Electron Microscopy (TEM)

Microscopic imaging using a transmission electron microscope was performed for visualizing the shape of the TBH-loaded particles. These images were also used for estimating the particle size of the formed particles. Additionally, TEM was also used to confirm if any aggregation of the polymeric particles occurred. Square copper grids with a thin (10 nm) film of carbon and 300 mesh (VWR, Radnor, PA, USA) were dipped in the suspensions and set aside to dry at room temperature, before imaging with a JEM 100CX TEM (Jeol Ltd., Tokyo, Japan).

### 2.9. Preparation of TBH Nano-Gel Formulation

Carbopol^®^ Ultrez-10 was slowly dispersed in the obtained nanoparticle suspension and allowed to hydrate with magnetic stirring at 500 rpm until a lump-free dispersion was achieved. The homogenous dispersion was neutralized with triethanolamine in an amount corresponding to a 1.5:1 ratio with respect to the mass of Carbopol^®^, as recommended by the manufacturer [45,46]. The resulting gel was gently but thoroughly manually mixed to avoid introducing air bubbles and to ensure uniform distribution of the neutralization agent. The concentration of the thickening agent in the obtained formulations was 0.5% (*w*/*v*). A control gel formulation with powdered TBH equivalent to the average drug content in the nanospheres dispersed into the carbomer dispersion was prepared as per the process described above.

### 2.10. Characterization of TBH Nano-Gel Formulation

The physicochemical characterization performed on the TBH nanosphere-loaded carbomer-based gels included the analysis of pH, viscosity, and content uniformity. The pH of the nano-gel was measured with a calibrated VWR sympHony™ B10P benchtop pH meter (VWR, Radnor, PA, USA) at room temperature.

The viscosity of the gel was measured using a Brookfield Viscometer DV3T-HB (Brookfield Engineering Laboratories, Middleboro, MA, USA) fitted with a cone spindle CPA-40Z. The measurement was conducted at 25 °C ± 0.5 °C with 0.5 mL of the gel loaded into the sample cup. The measurement speed was 1.5 rpm with multipoint sample collection every 10 s for 3 min. The reported viscosity value was the value recorded at the end condition.

Content uniformity analysis on the TBH nano-gel involved collection of three accurately weighed aliquots each from different locations of the nano-gel and control gel containers. The aliquots were dissolved in 5.0 mL of methanol by vortexing for 5 min followed by sonication for 10 min. The resulting solution was filtered through a 0.45 µm PTFE filter and quantified using the pre-validated HPLC method. Triplicate measurements were performed for each investigated formulation.

### 2.11. In Vitro Drug Release Study

The in vitro release testing (IVRT) of TBH from the nanosphere suspension and the nanosphere-loaded gels was studied using SnakeSkin Dialysis Tubing, 10K MWCO (Thermo Fisher Scientific, Waltham, MA, USA) sandwiched between the donor and receptor chambers of vertical Franz diffusion cells (Logan Instruments, Somerset, NJ, USA) with a receptor volume of 5 mL and an effective diffusion area of 0.64 cm^2^. The receptor compartment was filled with PBS (pH 7.4) containing 40% (*v*/*v*) ethanol to achieve sink conditions and contained a 3 mm magnetic stir bar for constant stirring of receptor media at 600 rpm. The assembled Franz cells were placed in FDC-24 heat blocks (Logan Instruments, Somerset, NJ, USA) set at 37 °C, and were allowed to equilibrate for 15 min before applying the formulations in the donor compartment. All formulations were applied in excess to achieve an infinite dose of TBH. Samples of 300 µL were withdrawn through the sampling arm of the Franz cells at predetermined time points followed by immediate replenishment with the same volume of fresh receptor media. The pre-validated HPLC method was used to determine the drug content of each sample, and cumulative drug contents were plotted against the time to obtain the release profile.

### 2.12. Ex Vivo Nail Permeation Study

#### 2.12.1. Preparation of Nails

Frozen human cadaver fingernail samples were obtained from Science Care, Phoenix, AZ, and were stored at −20 °C before use. Nail plates from index, middle and ring fingers of the donors were used because of their structural similarity. Nail plates were thawed at room temperature (25 ± 2 °C) and rinsed with DI water, cleaned by removal of adhering tissues with forceps, followed by another rinse with DPBS. The cleaned nails were then briefly immersed in 70% ethanol for disinfection, then dried with Kimwipes. Cleaned nails were visually inspected for any cracks or fractures and characterized in terms of weight and thickness to be selected for the permeation study. The selected nail plates were free from any visual cracks, were of similar weight, and were of 0.7 mm ± 0.15 mm in average thickness, measured at three different points using a digital micrometer (Mitutoyo, Kawasaki, Kanagawa, Japan).

#### 2.12.2. Pre-Treatment of Nails

A permeation enhancement pre-treatment of the nail plates was performed as a modification of previously demonstrated treatments [47,48]. An aqueous solution of 10% *w*/*v* TGA and 10% *w*/*v* PEG 400 was used for the pre-treatment of the nails, achieved by overnight incubation of the nail plates. After removal of the nails from the pre-treatment solution, they were dried and utilized for permeation testing.

#### 2.12.3. Nail Permeation Study

Modified vertical Franz diffusion cells clamped with Neoflon^®^ nail adapters (PermeGear, Hellertown, PA, USA) with an effective exchange area of 0.2 cm^2^ were used for in vitro permeation testing (IVPT) on the nail. The receptor chamber consisted of 5 mL of DPBS (pH 7.4) with 2% by weight of polysorbate 80, and 0.01% by weight of gentamicin, maintained at 37.0 ± 0.5 °C and stirred constantly at 600 rpm. An amount of 0.5 g of TBH nano-gel or control gel formulations were applied to the exposed portion of the dorsal surface of the nail plates mounted on the adapters. Samples of 300 µL were collected from the sampling arm of the Franz cells at 0 h and 12 h, followed by daily sampling for 15 days, and an equal volume of fresh receptor media was replenished after every sampling.

### 2.13. Analytical Method for Drug Determination

The content of TBH in the samples described above was quantified using an Agilent 1100 Series HPLC system with an autosampler combined with a UV detector, and a reverse phase Eclipse XDB-C-18 column (150 mm × 4.6 mm; 5 µm) (Agilent Technologies, Santa Clara, CA, USA) maintained at 40 °C. The method consisted of a gradient elution, with the mobile phase being acetonitrile and water, both containing 0.1% (*v*/*v*) formic acid starting at a ratio of 35:65 (*v*/*v*) and changing to 80:20 (*v*/*v*) over four minutes, maintained for the run time of 10 min, with a 2 min post run at a flow rate of 1 mL min^−1^. With an injection volume of 10 µL and UV-visible detection at 284 nm, the analyte peak was observed at 3.6 min. The method was validated with respect to linearity, precision, accuracy, and repeatability. Calibration curves were constructed with concentrations ranging from 25 µg/mL to 1000 µg/mL, as well as from 1.5 µg/mL to 100 µg/mL, and linear regression analyses showed a coefficient of correlation, R^2^ > 0.99 for both curves. The LOD and LOQ were 0.01 and 0.03 µg/mL, respectively.

### 2.14. Statistical Analysis

All statistical analysis of the DoE data, including the creation of experimental designs, as well as performing the power analysis of designs, was conducted using JMP^®^ Pro 15 by SAS Institute, Cary, NC. Regression analysis was utilized for prediction modeling, and a model was considered significant only if the p value was <0.05. The same threshold was used for the p value to confirm statistical significance of any comparison of groups, wherever applicable, which was performed using ANOVA. Correlation analyses were performed either using Agilent ChemStation B.04.03 or Microsoft Excel v. 16.47.1. The experimental data for drug quantification, recovery calculation and analytical method validation was obtained from triplicate measurements and reported as mean ± standard deviation.

## 3. Results and Discussion

### 3.1. Identification of QTPP, CQAs, CMAs and CPPs

The concept of QbD for formulation development relies on systematic approaches and rational scientific principles for achieving target quality in the end product [38]. The important steps involved in the development of TBH-loaded nanospheres based on the QbD approach are summarized in Figure 2.

The target quality of the TBH nanospheres was defined by several desired quality attributes, with the CQAs being a Z-ave of <250 nm, PDI of <0.5, and maximized recovery, as summarized in Table 5. ZP, particle aggregation and particle shape were analyzed for the finalized particles to confirm they fell within the desired quality range. These characteristics were selected on the basis of prior knowledge as well as review of published literature [49]. The rationale of forming nanospheres for TBH delivery is supported by the findings by Dhamoon et al. [50] that show the promise of nanoparticles in antifungal delivery—negligible side effects and improved drug release and penetration being the most important. It was also shown that extensive research is being undertaken towards novel options for nail delivery of antifungals. With respect to nail delivery, nanoparticles are known to improve solubility, bioavailability, efficacy, and stability of antifungal drugs, that are generally lipophilic [51]. The small size of the particles is believed to assist with accumulation of the drug into the pores and crevices of tissues, thus, facilitating availability of the drug at the desired site of action [43], and minimizing the systemic distribution of the drug [6]. The positive charge of the particles further assists with bioadhesion on and around nails, as skin contains negatively charged mucoproteins and the nails are formed mostly of keratin which is negatively charged [43].

Figure 3 is the Ishikawa diagram (also known as “fishbone” or “cause-effect” diagram) which helps highlight the various material and process attributes that could impact the quality of the nanospheres. The material-related factors that could influence product quality range from choice of drug and polymer, to their concentrations or ratios. Other material factors are surfactant in the external phase and its concentration. It is known that the volumes of the internal and external phases could also affect the properties of the particles [52]. Therefore, the process parameters involved in the various stages of nanosphere development were included, such as mixing the primary emulsion, ultrasonication parameters to create a finer emulsion, solvent evaporation of DCM to leave the nanospheres in the suspension, and even the ultracentrifugation step for washing the particles that alter the final quality of the formed nanospheres. It also captured the conditions of the preparation that could affect product quality, along with the characteristics that needed to be studied to confirm that the desired nanospheres were achieved.

The clarity obtained from the cause–effect diagram, when combined with prior knowledge or experience about the process, enabled the categorization of risk from all the parameters involved in the formation of the product. The categorization was performed at three levels and all factors were assigned a low, medium, or high risk. Table 6 and Table 7 show the risk assessment with the nanosphere CQAs against material and process variables, respectively, to define CMAs and CPPs based on understood risk-level. The objective of such a risk-based approach was to perform experimentation and use existing knowledge to achieve a design space within which the risk-level from these variables was minimized, and product quality could be expected to be of a desired level. The assessment combined with the initial DoE resulted in the identification of PVA concentration in the external phase, as well the ratio of the drug to polymer, as the most critical variables for nanosphere quality. For assuring important quality characteristics and process repeatability, mixing and sonication parameters were fixed to the values mentioned in Table 4.

Justification for the assigned risk to some of the parameters was supported in several literature summaries, specifically in application to nanocarriers for drug delivery. The type or properties of the drug were assigned as low risk on particle size and PDI for instance, as it was dissolved in the organic phase polymeric solution [49]. As another example, the polymer type and concentration were high risk factors for zeta potential, because the polymer forming the nanosphere carried a charge, and would impart that charge to the surface of particles.

### 3.2. Preliminary and Secondary Screening DoE

The early exploratory designs were set up with the objective of confirming the independent variables and their ranges as well as for the selection of the polymer to form the optimized nanospheres. Performing screening experimentation for better understanding of the product development process is good practice, since it assists in generating a final design with the most relevant parameters [35,49,53]. DoE methodology can be cost effective, time saving, and can result in an optimum quality product as compared with the one factor at a time (OFAT) approach, by statistically enabling the establishment of a design space for a quality product [54].

The preliminary screening design shown in Table 8 contained two responses (Z-ave and PDI) with identical 2^4^ full factorial designs for two polymers—EC and E-RSPO—consisting of two levels each of PVA concentration, drug to polymer ratio, mixing time, and sonication time as independent variables. ANOVA was applied to the responses obtained from the trials, using least squares fit analysis to determine the statistical significance and magnitude of impacts from the main effects, as well as interaction effects of the variables. The results showed that at the levels selected for this experimental design, Z-ave and PDI were both significantly impacted only by the PVA concentration with *p*-values of 0.0005 and 0.009, respectively, when nanospheres were made with EC. For nanospheres made with E-RSPO, the impact of any of the variables on both responses did not appear to be significantly different from the others. This was because most of the trials with the E-RSPO resulted in consistently small particles with acceptably low polydispersity at the set variable levels. This was not the case with the EC particles, which had more scattered response ranges with a relatively higher number of particles above the acceptable limit of 250 nm. It could be that EC particles show a higher tendency to agglomerate than E-RSPO, as the latter carries a charge which contributes to repulsion between other like-charged particles. Both polymers were included in the secondary screening experimental trials to be certain that the choice of the polymer was justified. The initial screening DoEs could be used in different ways to obtain the required information for the final optimization DoE. Raina et al. used such screening experiments to select the right lipid and surfactants to be used in the 3^2^ full factorial final design, in which they varied the amount of the surfactant and ratio of solvents in organic phase [55]. This gave the required information about the process to decide which factors needed to be optimized further to achieve the desired product quality.

The results from the preliminary set of experiments showed that within the ranges tested, mixing and sonication times did not have a statistically significant impact on the responses, yet these were important parameters needed to achieve the desired particles. To ensure that well dispersed suspensions of uniformly sized particles were achieved in further experimental trials, the mixing time was fixed to 10 min and total ultrasonication time was fixed to 15 min. This lowered the initially accessed risk-levels from these parameters, as the fixed levels were shown to achieve their intended effects. To further reduce the risk from parameters and to reach to an improved experimental design, as well as to understand the impact of phase volumes in the emulsion, the secondary experimental design consisted of a wider range of PVA concentration (0.25% *w*/*v*, 0.5% *w*/*v* and 1% *w*/*v*) and drug to polymer ratio (1:1, 1:2, and 3:1) at three levels each, as well as three levels of external phase volume to internal phase volume ratio (10:2, 10:4, 10:6). The ratios of the phase volumes were changed by keeping the EPV fixed to 10 mL, and varying the IPV to 2 mL, 4 mL, and 6 mL. As the number of levels of the variables increased, the DoE approach was modified to a response surface design. Modifications of the traditional RSM design were conducted to minimize the size of the experimental design, which could become very large with three factors at three levels, while achieving meaningful analysis [56]. The design resulted in 13 randomized experimental trials, shown in Table 9. The responses were still limited to Z-ave and PDI for the sake of keeping the analysis relatively less complicated. The results from these experiments helped provide useful information to narrow down the parameters and obtain a refined final experimental design.

From the results of the secondary design, it was observed that the EPV to IPV ratio of 10:2 showed particles in acceptable ranges most of the time, however, a statistical impact was not seen from this parameter. Therefore, EPV was fixed to 10 mL and the IPV to 2 mL, for further experiments. There were some EC particles that were too big to be measured accurately, with a total of seven samples with Z-ave values outside the acceptable quality. With E-RSPO, on the other hand, only four resulting particles were unacceptably large, all being measurable, and three PDI values were above the desired limit. Upon measuring the particle sizes again after leaving the suspension at ambient conditions for 3 days (data not shown), it was observed that the EC particles showed an even further increase in particle size, and the E-RSPO particles did not. As a result, E-RSPO was selected as the polymer of choice to form the nanospheres. E-RSPO has been researched for topical TBH delivery before, and has been shown to be a compatible, film forming and bioadhesive polymer option [57]. It has been reported by Mahaparale et al. in the past that microsponges formed with Eudragit^®^ polymers were not spherical and rigid, and better particles were obtained with EC [58]. Therefore, it was important for us to verify the nanosphere morphology using microscopy.

### 3.3. Final Experimental Design

Based on the results from the initial experimental trials, the two independent variables selected for the final refined DoE were PVA concentration and drug to polymer ratio; and other material and process parameters were assigned fixed values, as shown in Table 4. The levels at which PVA concentration and drug to polymer ratio were varied were spread even further to PVA concentrations of 0.25% *w*/*v* (low), 1% *w*/*v* (medium) and 2% *w*/*v* (high); and the drug to polymer ratio of 1:4 (low), 1:1 (medium) and 4:1 (high). The dependent variables that were measured for the experimental trials in the 3^2^ design were Z-ave, PDI, and recovery.

A statistical analysis of the power of an experimental design can be used in advance to estimate the sensitivity of the design to detect a real effect from the variables [59]. With a 3^2^ full factorial design, run in triplicate, a 27-run design was formed. A power analysis could help determine if a DoE with lesser trials could still be capable of performing the desired analysis. The power of a DoE should be at least 80% to detect parameter effects [59]. The number of replicates needed to achieve a power of over 80% increases rapidly, but there is little advantage of this additional power [60]. A power analysis was performed to predict the confidence of analysis with lower run designs obtained by varying trial replicates and number of center points. Table 3(b) shows the values for experimental trials, replicates and center points used to perform the power analysis. Figure 4 shows the outcome of the power analysis on experimental designs of different trial runs, replicates, and center points.

The smallest design in the comparison was the 18-run design that resulted from two replicates and no center points. The designs with 20, 22, and 24 trials were all formed of two replicates as well, but had one, two, and three center points, respectively. As already mentioned before, the 27-run design resulted from triplicates without any center points. As seen in Figure 4, the 18-run design had a power of 79.7% for the interaction effect of the independent variables, which was just under the 80% threshold. The power for all other terms analyzed with the designs was above 80%. The difference between power levels of the designs was not significant as can be clearly seen in the graphical plots, except for the 27-run design. Based on this analysis, the 20-run design was selected for the further optimization and randomized order of experimental trials that resulted, as shown in Table 10. The CQAs shown under the response side of the table did not include zeta potential, as desirable responses were expected based on our prior experience with the polymer at the utilized levels. In our previously published work utilizing QbD for the development of a niosome-based formulation for topical delivery, we utilized a 2^5^ DoE consisting of 32 trials in the design [61]. This present study is an attempt at refinement of the design to obtain a relatively simpler design with comparable efficiency.

### 3.4. Response Analysis

The response values obtained from the experiments as shown in Table 10, were used to fit against the model as shown in Equation (1), using regression analysis, and ANOVA was utilized to test the fit. With the alpha value for the analysis set to 0.05, the results summary showed a significant impact from the main effects as well as the interaction effect of the independent variables. The overall effect summary is shown in Table 11 by the LogWorth of each model effect, which was defined as −log_10_(*p*-value). PVA concentration appeared to have the highest effect on nanosphere CQAs, and the drug to polymer ratio had an effect lower than the interaction effect of the two variables.

The desirability assigned for recovery was to maximize the response; and for Z-ave and PDI, to minimize the response value. The prediction profiler function in JMP^®^ which predicts the effect of the changes in independent variables on the dependent variables could be used to reflect the optimized combination of parameters by maximizing desirability, as shown in Figure 5.

The effect leverage plots for all three responses are shown in Figure 6, reflecting the fit of each term to the model. They were based on the hypothesis that the effect is not in the model, given all other effects are in model. The red line is line of fit of least squares, and the shaded red bands represent the confidence curves. The blue horizontal line signifies the hypothesis, the effect is considered significant if the confidence curves cross the horizontal line, and not significant if the confidence curves contain it.

From the leverage plots above and the regression analysis, it can be said that the data fit the model well. All effects were statistically significant, as can be seen from the p-values.

#### 3.4.1. Effect on Particle Size

The particle size of the nanospheres ranged between 101.4 nm (F4) and 197.7 nm (F15), and was impacted significantly (*p* < 0.05) by variations to both main effects as well the interaction effect of the independent variables. The mathematical expression of the regression explains the parameter effects on Z-ave as shown below in Equation (3):(3)$$Z–\mathrm{ave}=128.151-26.813\left[\frac{\left(\mathrm{PVA}\;\mathrm{Conc}.-1.125\right)}{0.875}\right]-14.284\left[\frac{\left(\mathrm{Drug}/\mathrm{Polymer}-2.125\right)}{1.875}\right]\;+\left[9.829\left\{\frac{\left(\mathrm{PVA}\;\mathrm{Conc}.-1.125\right)}{0.875}\right\}\left\{\frac{\left(\mathrm{Drug}/\mathrm{Polymer}-2.125\right)}{1.875}\right\}\right]$$

The positive sign in the equation indicates a synergistic effect and negative signs indicate an antagonistic effect from the parameter the sign precedes. As the PVA concentration increased, the particle size decreased. Similarly, the Z-ave decreased with a decrease in the drug to polymer ratio. However, from the prediction profiler in Figure 5 and the surface plots in Figure 7, it can be seen that the impact from the drug to polymer ratio and the combination effect on Z-ave was not as high as compared with the PVA concentration. This is also seen from the higher coefficient value for PVA concentration in Equation (3).

The purpose of PVA in the external phase is to act as an emulsifying agent or stabilizer for the emulsion. It can exhibit surfactant-like properties and aids in reduction in interfacial tension between the organic and aqueous phases. This explains why an increase in the concentration of this stabilizer in the aqueous phase reduces the size of the emulsion droplets and hence, particles formed. It also serves the purpose of preventing particle aggregation as the solvent evaporates by avoiding coalescence of droplets by means of steric stabilization [52]. It has been generally seen that with an increase in concentration of the polymer forming the nanoparticles, the particle size increases [52,62]. It also happens that beyond a certain threshold, if the polymer concentration is increased further, the particles could form aggregates [62]. In TBH nanospheres, it was found that at the levels at which the polymer amount was varied, in combination with the other conditions of the experiment, the process of ultrasonication was well capable of efficiently forming particles under the size of 250 nm, as desired.

#### 3.4.2. Effect on PDI

The range of PDI observed in the nanospheres, as seen in Table 10, varied from 0.054 to 0.254. All these particles were in the desired quality range of <0.3. A PDI of higher than 0.3 signified a broad distribution of the particles and was not ideal as it could mean lower stability of the nanoparticulate suspension [52]. The effect test on PDI showed that the main effects of the independent variables did not have a significant impact on the product quality. However, the interaction effect of the two variables showed a significant impact with a *p*-value of 0.0002. The PDI data fit to the model can be seen in Equation (4).
(4)$$\mathrm{PDI}=0.115+0.019\;\left[\frac{\left(\mathrm{PVA}\;\mathrm{Conc}.-1.125\right)}{0.875}0.875\right]+0.013\;\left[\frac{\left(\mathrm{Drug}/\mathrm{Polymer}-2.125\right)}{1.875}\right]\;+\left[0.058\left\{\frac{\left(\mathrm{PVA}\;\mathrm{Conc}.-1.125\right)}{0.875}\right\}\left\{\frac{\left(\mathrm{Drug}/\mathrm{Polymer}-2.125\right)}{1.875}\right\}\;\right]$$

The highest coefficient can be seen with the interaction term, and the effect was synergistic. The PDI of the nanosphere suspension obtained by the formulation process showed little variability and fell within acceptance criteria. The PVA concentration and drug to polymer ratio, at the levels tested, did not significantly impact PDI, but the interaction from their combination did, as can be seen in the surface plot in Figure 7.

#### 3.4.3. Effect on Recovery

The drug recovery of nanoparticles is an important characteristic as it determines the amount of drug that will be available for delivery. Since most topical fungal treatments are relatively long in duration, it was desired that recovery be maximized in order to reduce the number of applications required, as well as to allow for a sustained release of the drug from the site where the nanospheres are deposited in the skin or nails [43,50]. The observed values from Table 10 show recovery ranging from as low as 11.97%, to as high as 60.22%. Equation (5) is the regression equation obtained for drug recovery.
(5)$$\mathrm{Recovery}=33.850+8.496\;\left[\frac{\left(\mathrm{PVA}\;\mathrm{Conc}.-1.125\right)}{0.875}\right]+1.716\;\left[\frac{\left(\mathrm{Drug}/\mathrm{Polymer}-2.125\right)}{1.875}\right]\;-\left[9.555\left\{\frac{\left(\mathrm{PVA}\;\mathrm{Conc}.-1.125\right)}{0.875}\right\}\left\{\frac{\left(\mathrm{Drug}/\mathrm{Polymer}-2.125\right)}{1.875}\right\}\right]$$

The coefficient values show that PVA concentration, and interaction effect of PVA and drug to polymer ratio, have higher impacts on the recovery. The statistical effect test confirmed this, with *p*-values of 0.0008 for these two factors, and showed that the main effect from drug to polymer alone was insignificant (*p* = 0.3815). The surface plot in Figure 7 for recovery shows the intensity rising at the highest PVA concentration. Therefore, the recovery of TBH nanospheres can be said to be increasing with increasing concentration of PVA in the aqueous phase. The surface stabilization effect PVA can be attributed to this synergistic effect on recovery.

### 3.5. Model Validation

The optimized formulation with maximum desirability predicted by the model, as shown in Figure 5, was a combination of 2% *w*/*v* PVA and 1:4 ratio of drug to polymer. According to the model, the resulting formulation with this combination would have an average particle size of 105.8 nm, with the range being from 90.8 nm to 120.8 nm, a predicted PDI range from 0.023 to 0.103, and recovery of 42.51% to 57.86%. To validate the model and verify the prediction accuracy, the formulation with 2% *w*/*v* PVA and 1:4 drug to polymer ratio was reformulated in triplicate, and the mean values obtained for the CQAs were compared with the mean model predicted values, as shown in Table 12.

As seen from the *p*-values in Table 12, no significant difference was seen between the two groups, thus, validating the model and its prediction accuracy. It can, therefore, be concluded that the model generated from the experimental design can accurately estimate nanosphere CQAs when the parameters are varied within the design space.

### 3.6. ZP of TBH Nanospheres

The ZP of the nanospheres was identified to be an important quality attribute because it plays a major role in their stabilization in suspension. A high ZP provides repulsive forces between similarly charged particles, and is supplemental to the stability impacted by the surface active stabilizer [61,63]. The ZP of the optimum nanosphere formulation was measured to assess this potential electrostatic stabilization of the investigated system. The average ZP of these particles was found to be +43.5 mV, which fitted the desired QTPP for the nanospheres (Table 5). In addition to the stability, having a positive charge over 40 mV was desired for the intended topical antifungal delivery, as these particles are expected to be bioadhesive to the skin and nail tissues, as skin contains negatively charged mucoproteins, and the nails are mostly formed of keratin which is negatively charged [43]. By means of adhesive forces, the particles are expected to be able to be retained in superficial layers of the tissue, and release the drug over an extended period.

### 3.7. DSC Analysis

The curves obtained in DSC experiments for the pure active ingredient, polymer, physical mixtures of the drug and polymer, and for the investigated nanospheres, are depicted in Figure 8. The curve recorded for pure TBH shows a sharp endothermic peak at 210.2 °C, which corresponded to the melting point reported in the literature [64,65]. The shape of the peak suggests possible decomposition observed immediately after the melting process, which was also indicated by Kuminek et al. [66]. TBH melting peaks are also observed in physical mixtures containing different amounts of the active ingredient. However, in the case of drug/polymer mixtures at 1:1 and 1:4 ratios, the peaks shifted slightly to lower temperatures. This shifting of the melting peaks might be indicative of a possible interaction between the components of the particles. In the curve recorded for pure polymer, glass transition is visible at about 60 °C, which was already reported by Fujimori et al. [67]. Glass transition temperatures are also discernible in the curves obtained for drug/polymer mixtures at 1:1 and 1:4 ratios. The lack of endothermic peaks corresponding to melting process indicates that the polymer revealed purely amorphous properties. In the plots recorded for the investigated optimized nanosphere formulation, no endothermic peak related to TBH melting is visible, indicating that the drug was incorporated in the particles in amorphous form, which is favorable in terms of dissolution [68] and topical delivery.

### 3.8. Nanosphere Morphology

Morphological assessment of the nanospheres was conducted using TEM, and the shape of the particles could be confirmed as spherical, as shown in Figure 9. TEM analysis could also be helpful in several other important characteristics, such as visual estimation of the particle size and distribution, presence of drug crystals free from the nanospheres, as well as aggregation of the particles [69]. The images showed distinct spherical particles with particle sizes conforming to the range obtained by particle size analysis by DLS. It should be noted that difference in the size of the particles as measured using DLS and TEM may be expected, due to the sample preparation procedure differences [55] and the principles of these techniques—DLS measured the hydrodynamic radius in suspension and TEM analyzed dried particles. Particle aggregation or drug crystals were not seen in the images of the optimized nanospheres. It was also important to confirm the shape of the particles as there was evidence in the literature to doubt it, but that could have been a result of unique experimental conditions in the prior work [58].

Colloidal particles have been explored for decades as enhanced drug delivery systems, and such novel antifungal formulations for topical dermal and ungual delivery have lately been of interest [12]. Nanocarrier-based topical preparations exhibit specific advantages such as targeted delivery of drugs that put conventional methods at a loss. Nanovesicular terbinafine systems by Elsherif et al. [64] and liposomal formulations of novel antifungals by Naumann et al. [70] are relatively recent examples demonstrating the sustained release of topical formulations, important since these treatments often demand long-term treatments [71]. Small particle size is one of the most important characteristics of such formulations, as it is favorable for permeability and could assist in improved drug deposition in the superficial layers [72]. Drug-containing particles deposited in the layers, folds and cavities of the tissue could achieve long-term and effective treatments on the skin and mucosa, while avoiding systemic exposure with potentially harmful drugs [73,74].

### 3.9. Characterization of TBH Nano-Gel Formulation

The prepared TBH nano-gels were characterized for pH, viscosity, and content uniformity to ensure the quality of the formulation as desired. The pH of the nano-gel was found to be 6.9 ± 0.2, and that of the control gel was 6.6 ± 0.3. This means that the gel drug product would not be irritant to the surrounding tissue around the nails upon contact. Another advantage of having a relatively higher pH is that the solubility of TBH increases at lower pH values, as shown in literature [75,76], and such a condition may favor partitioning of the drug into the vehicle before it could be delivered as nanospheres. Additionally, at lower pH values, terbinafine becomes ionized, and has been shown to bind to the nail keratin [77]. Removal of the drug from the nanospheres and solubilization in the hydrogel will also cause the ionized terbinafine to gain molecular weight and become less permeable. Moreover, the nail plate, formed of keratins containing disulfide linkages, has an isoelectric point (pI) of about 4.0–5.0 and carries a net negative charge at physiological pH (pH 7.4) [77,78], making the conditions favorable for the positively charged nanospheres to adhere to the nail, also in the deeper layers due to its low particle size, and this would lead to a more efficient drug delivery system.

The mean viscosity of the TBH nano-gels was found to be 37,320 cP ± 725 cP, and the control gel had a viscosity of 35,300 cP ± 555 cP. This is high viscosity, and desirable because it would support a longer residence of the gel on the nail upon application, instead of flowing down rapidly. A robust formulation such as the one achieved will be able to provide sufficient time for the hydrogel to stay on the nail plate and help achieve a hydrated state, making it more permeable while also allowing for delivery of the nanospheres from the gel formulation into the nail plate [77,79].

The concentration of TBH (% *w*/*w*) per unit weight of the gels was found to be 0.10% ± 0.02%, and 0.10% ± 0.01%, for the nano-gel and control gel, respectively. The low relative standard deviations in both test and control gel samples collected from different locations of the bulk of the gel confirmed their content uniformity, which validated the formulation preparation process for producing a consistent and uniform gel. This also verified that the amount of active ingredient being applied during the release and permeation studies was comparable in the test and control formulations. The characterization of TBH nano-gel confirmed that the product quality was adequate and met the desired quality.

### 3.10. In Vitro Drug Release Study

The cumulative release of TBH from the control and nano-gel formulations is plotted against time in Figure 10a; it can be noted that the cumulative amount of drug released from the two types of gels was similar at the end of the 24-h IVRT study. However, the drug release profiles show that the control gel showed a higher release early on in the study, and started to plateau towards the end; whereas the TBH nano-gel showed a controlled release over time and appeared to gradually build up the drug load in the receptor media, and did not appear to have plateaued at 24 h. The drug release profile from TBH nano-gel was found to obey Higuchi’s square root diffusion model, as shown in Figure 10b, with a correlation coefficient (r^2^) of 0.99 [80,81]. The data fitting was based on Equation (6) as described by the Higuchi model:(6)$$Q=K_{H}\times \surd t$$
where *Q* is the amount of drug released in time *t*, and *K_H_* is the Higuchi constant. This model describes the drug release from insoluble matrix and can apply to controlled release systems. On the other hand, the release of TBH from the control gel did not fit the Higuchi model and did not follow controlled release such as the nano-gel. A controlled drug release profile is desirable for the intended application as onychomycosis treatments are long-term due to the limited permeability and slow nail growth [71]. A sustained delivery of the active ingredient is needed to maintain antifungal activity at the site of infection. Similar research such as nanovesicular terbinafine systems by Elsherif et al. [64] and liposomal formulations of novel antifungals by Naumann et al. [70] are relatively recent examples demonstrating use of controlled or sustained drug releasing topical formulations.

### 3.11. Ex Vivo Nail Permeation

Conducting nail permeation studies on human cadaver nails mounted on modified Franz diffusion cells can help in the understanding of the applicability of the drug delivery system in actual human use. Figure 11 demonstrates the drug permeation profile across nail plates achieved with the control gel and nano-gel after topical application, over 15 days. In contrast to the drug release comparison of the two formulations, the TBH nano-gel appeared to deliver the active ingredient across the nail plate into the receptor media faster than the control gel. As shown in the figure, TBH was detected in the receptor media on day 2 in the case of the nano-gel, whereas it was not seen until day 5 in the case of the control gel. Multiple applications of the nano-gel in a clinical setting could probably achieve a build-up of the active ingredient at the site of infection, as the gel cannot be expected to stay on the nails for 15 days. Nevertheless, the particle size, morphology, and surface charge of the of the nanospheres, along with the physical characteristics of the hydrogel, support the delivery of TBH during the initial hours after application of the nano-gel, which was lacking in the control gel that only had the delivery vehicle without the other favorable and necessary characteristics.

The flux of transungual drug permeation calculated from the linear portions showed that the nano-gel achieved a more than two-fold higher flux (0.0136 μg/cm^2^/h) compared with the control gel (0.0062 μg/cm^2^/h). The positive 40 mV surface charge of the nanospheres along with their small size could explain the faster and more efficient delivery of the drug into the negatively charged nail plate.

The pretreatment of the nails performed in this study is a modification of existing methods found in the literature which use a single permeation enhancer in the pretreatment solution [48,82]. We used a multi-mechanism permeation enhancing pretreatment approach containing TGA and PEG 400, both having a demonstrated ability to enhance the permeability of nails [83,84]. TGA is a thiol permeation enhancer known to disrupt the disulfide bridges linking keratin molecules, and PEG 400 is a humectant which can cause hydration and swelling of the nail plate [85]. We believed that the dual enhancer pretreatment strategy could achieve better permeation enhancement of the nail plate by hydrating and swelling the nail plates and simultaneously disrupting disulfide linkages at a deeper level, because TGA could reach more deeply into the swollen nail plate. To our knowledge, such a pretreatment approach has not been utilized in the existing research on transungual drug delivery.

The drug release and nail permeation studies confirmed that the TBH nano-gel formulation was more efficient at delivering the antifungal drug to the nail in a controlled manner, compared with the control gel. The successful delivery of terbinafine from the current delivery system can be attributed to multiple favorable factors, contributing simultaneously, in this system—favorable drug substance (molecular size, pK), colloidal drug nano-carrier (positively charged polymeric nanospheres of less than 200 nm particle size), formulation characteristics (pH above the isoelectric point of the nail and the pKa of the drug)—allowing the delivery of unionized terbinafine into and through the nail plate. Subsequent application of this TBH nano-gel could result in buildup of the nanoparticles and, therefore, the drug concentration in the nails, which could achieve even better permeation.

## 4. Conclusions

In this work, spherical nano-sized polymeric particles loaded with terbinafine hydrochloride were systematically synthesized by an understanding of the process with a risk-based approach. The optimum nanospheres as per the quality target product profile were formed with 2% *w*/*v* PVA in the aqueous phase as a stabilizer, and a 1:4 ratio of TBH and E-RSPO in the organic phase. These parameters, combined with a robust and repeatable formulation process, led to the development of nanospheres with the desired critical quality attributes of particle size, polydispersity, recovery, and zeta potential. The achieved nanospheres were loaded into a polymeric hydrogel that was conducive of effective ungual drug delivery. A modified multi-mechanism permeation enhancing pretreatment step was described in this research, that could be helpful in all types of future topical delivery approaches. Regular re-application with TBH nano-gel could potentially provide effective topical antifungal monotherapy for the treatment of onychomycosis. Further work is needed to analyze the effect of multiple applications of the formulation on the nail plate, and to check for antifungal effectiveness.

## Figures and Tables

**Figure 1 pharmaceutics-14-02170-f001:**
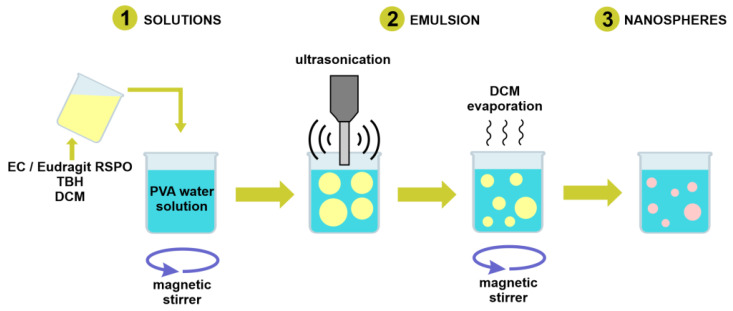
Schematic demonstrating the basic steps of nanosphere formation procedure.

**Figure 2 pharmaceutics-14-02170-f002:**
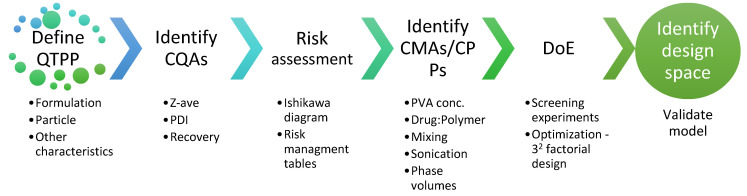
Elements and steps of the QbD approach for developing TBH nanospheres.

**Figure 3 pharmaceutics-14-02170-f003:**
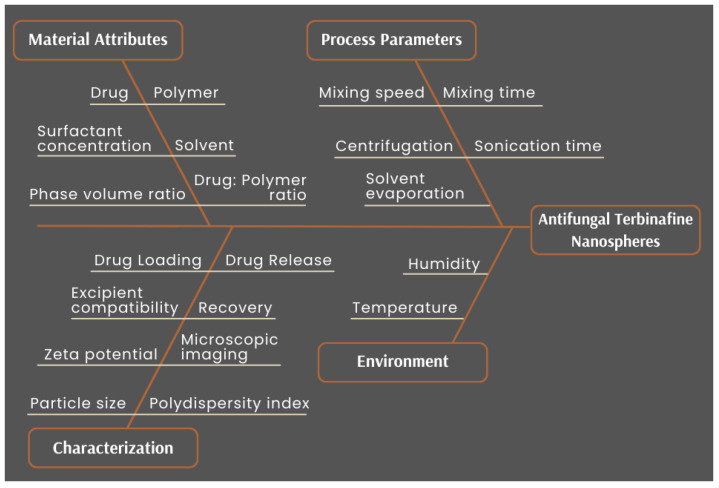
Ishikawa diagram for cause–effect mapping for the development of TBH nanospheres showing all factors that could impact product quality.

**Figure 4 pharmaceutics-14-02170-f004:**
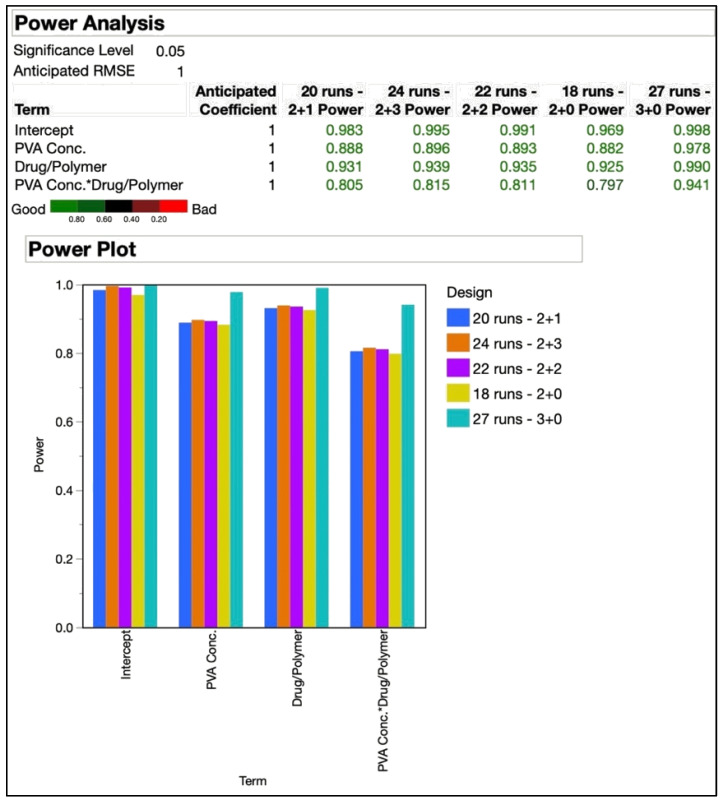
Power analysis of experimental designs with 18, 20, 22, 24, and 27 runs. The addition of the two numbers shown after number of runs represent the “number of replicates + center points”.

**Figure 5 pharmaceutics-14-02170-f005:**
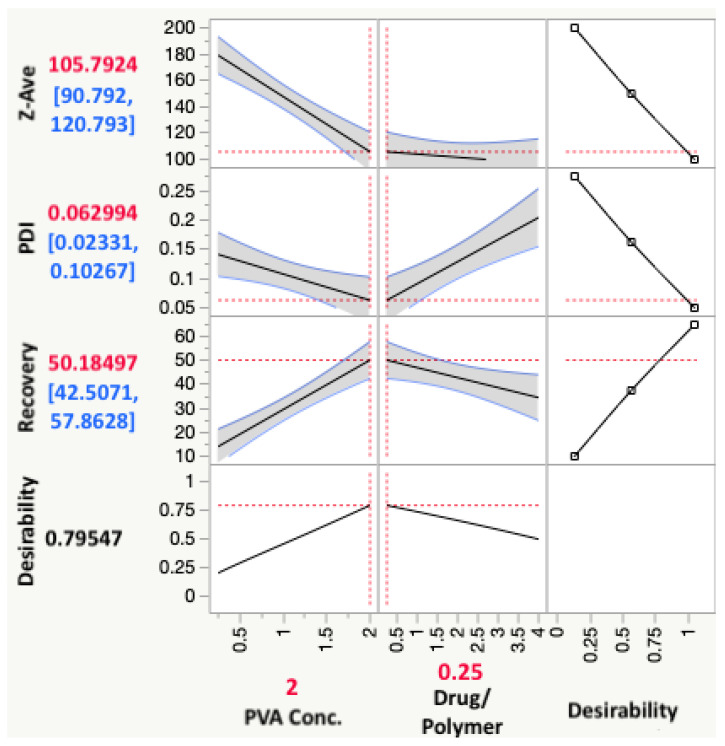
Prediction profiler from JMP^®^ showing the optimum parameter combination for maximized desirability, as well as the predicted values with confidence intervals of the responses expected.

**Figure 6 pharmaceutics-14-02170-f006:**
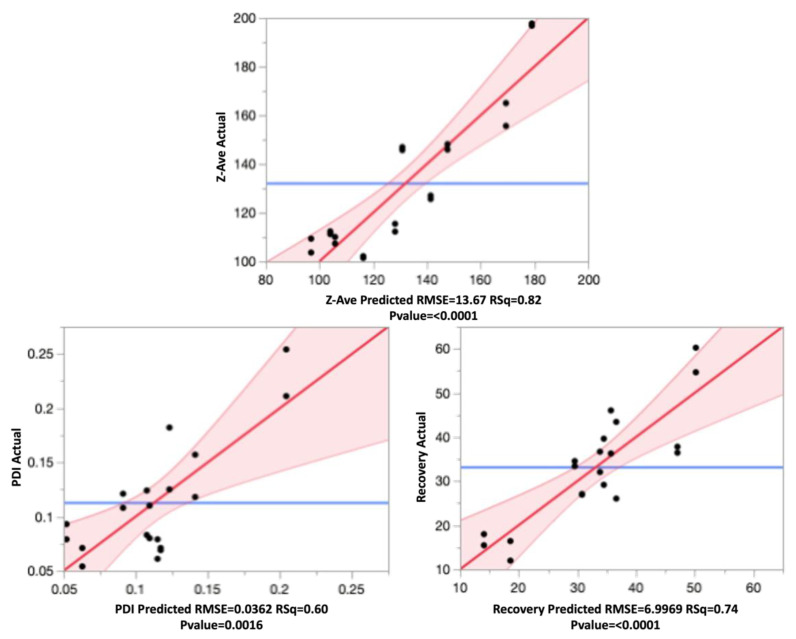
Leverage plots for Z-ave, PDI, and recovery showing the model without the term (blue line), model with the term (red line), and the confidence band (shaded red).

**Figure 7 pharmaceutics-14-02170-f007:**
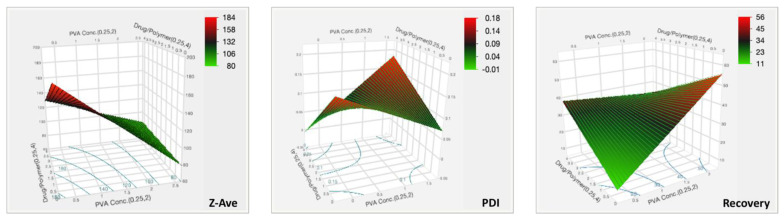
Three-dimensional surface plots for the CQAs of TBH nanospheres showing effect of variable parameters.

**Figure 8 pharmaceutics-14-02170-f008:**
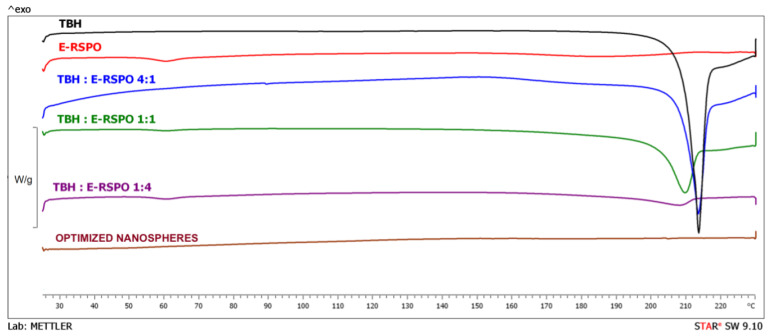
DSC curves obtained for the optimized nanospheres, pure initial substances and physical mixtures of the drug and polymer.

**Figure 9 pharmaceutics-14-02170-f009:**
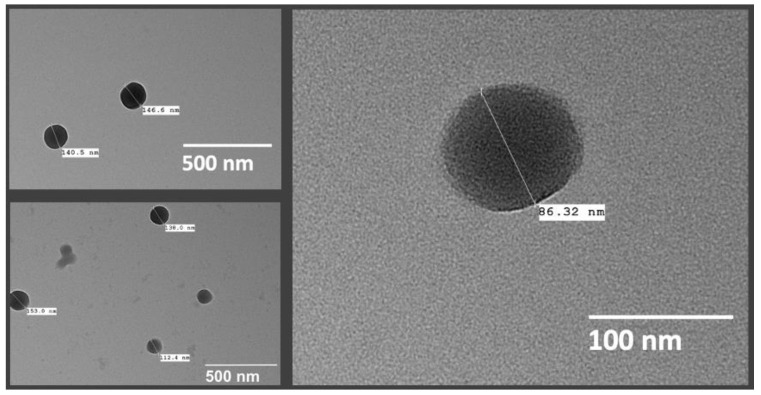
TEM images of TBH nanospheres.

**Figure 10 pharmaceutics-14-02170-f010:**
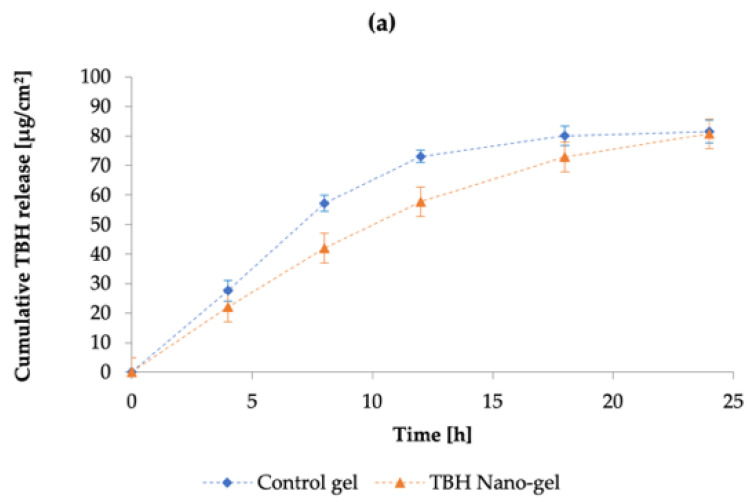
(**a**) The cumulative amount of TBH released per cm^2^ per unit time through the control and nano-gel formulations; (**b**) drug release kinetics of TBH from the control and nano-gel formulations as fitted to the Higuchi mathematical model, showing percentage of drug release as a function of square root of time. Blue diamonds (

) and orange triangles (

) represent the control and nano-gel, respectively.

**Figure 11 pharmaceutics-14-02170-f011:**
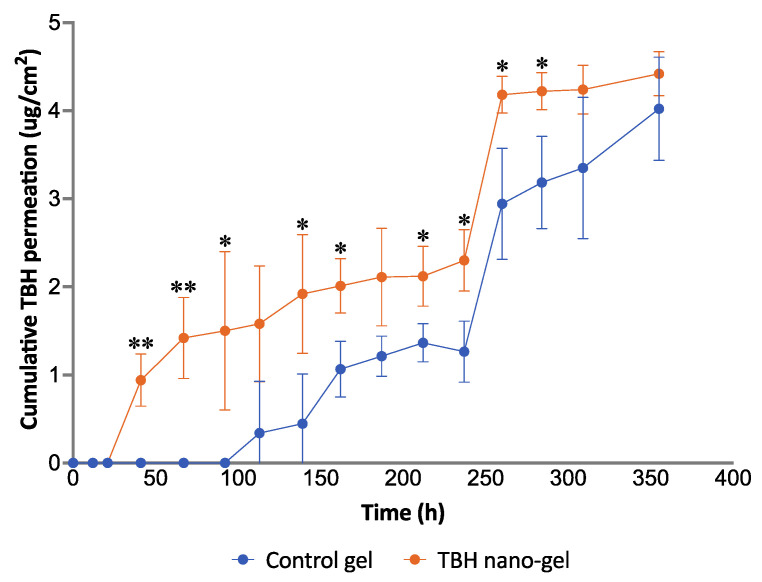
Ex vivo permeation study of control and TBH nano-gel formulations on human cadaver nail plates showing the cumulative amount of TBH permeated per unit area of the nail plate over time (* signifies *p*-value ≤0.05; ** signifies *p*-value ≤0.006).

**Table 1 pharmaceutics-14-02170-t001:** The levels of the independent variables used for the primary 24 full factorial design.

Independent Variables		Low (−)	High (+)
**CMAs**	PVA concentration (% *w*/*v*)	0.25	1
	Drug/polymer	1:1	1:2
**CPPs**	Mixing time (min)	0	10
	Sonication time (min)	5	15

**Table 2 pharmaceutics-14-02170-t002:** The independent variables and their values used for the secondary RSM-guided experimental design.

Independent Variables	Low (−1)	Medium (0)	High (+1)
**PVA concentration (% *w*/*v*)**	0.25	0.5	1
**Drug/polymer**	1:1	1:4	4:1
**EPV/IPV**	10:4	10:2	10:6

**Table 3 pharmaceutics-14-02170-t003:** (a) Independent variables and responses used for the final 3^2^ design; and (b) power analysis parameters.

Independent Variables	Low (−1)	Med. (0)	High (+1)	Exp. Runs	Replicates (N)	Center Points
**PVA concentration (% *w*/*v*) (X_1_)**	0.25	1	2	18	2	0
**Drug/polymer (X_2_)**	1:4	1:1	4:1	20	2	1
**Responses**				22	2	2
**Z-ave (nm)** **(Y_1_)**	**PDI** **(Y_2_)**	**Recovery (%)** **(Y_3_)**		24	2	3
				27	3	0
**(a)**				**(b)**		

**Table 4 pharmaceutics-14-02170-t004:** Constant parameters for the final 32 design and the values for each parameter.

Parameter	Amount
Drug and polymer total weight	100 mg
Internal phase volume	2 mL
External phase volume	10 mL
Mixing time	10 min
Mixing speed	500 rpm
Sonication time	0.7 s ON/0.3 s OFF, for 15 min ON time
Sonication power	70%
Vial	20 mL glass scintillation

**Table 5 pharmaceutics-14-02170-t005:** The target quality product profile of the nanospheres along with the critical quality attributes.

QTPP Elements	Target	CQA *	Justification
**Dosage form**	Polymeric nanospheres		To improve permeability, stability, and efficacy of API
**Route of administration**	Topical		Self-administrable, local drug delivery while avoiding systemic adverse effects
**Particle size (Z-ave)**	<250 nm	✓	Suitable for effective permeability
**Polydispersity index (PDI)**	<0.3	✓	Impacts physical stability and drug uniformity
**Recovery**	Maximum possible	✓	Ensures formulation efficiency and supports the desired drug release
**Zeta potential (ZP)**	>40 mV	✓	Helps with dispersion stability and particle uniformity
**Particle aggregation**	No visible signs of aggregation		Impacts permeability and stability
**Particle shape**	Spherical		Supports particle uniformity and allows penetration through narrow channels

* CQAs checked in the table are expected to vary with the parameters varied during development.

**Table 6 pharmaceutics-14-02170-t006:** Risk assessment of the material attributes on nanosphere CQAs.

CQAs	Material Attributes					
	Drug Type	Polymer Type	Drug/Polymer Ratio	Surfactant Type	Surfactant Conc.	Phase Volume Ratio
Average particle size	Low	Medium	High	Medium	High	Medium
Polydispersity index	Low	Medium	High	Medium	High	Medium
Recovery	Medium	Medium	High	Medium	High	Medium
Zeta potential	Low	High	High	Medium	Low	Low

**Table 7 pharmaceutics-14-02170-t007:** Risk assessment of the process parameters on nanosphere CQAs.

CQAs	Process Parameters							
	Solvent Evaporation	Mixing Speed	Mixing Time	Sonication Time	Sonication Power	Centrifugation Speed	Centrifugation Time	Centrifugation Temperature
Average particle size	Low	Medium	Low	Medium	Medium	Low	Low	Low
Polydispersity index	Low	Medium	Low	Medium	Medium	Low	Low	Low
Recovery	Low	Low	Low	Medium	Medium	Low	Low	Low
Zeta potential	Low	Low	Low	Low	Low	Low	Low	Low

**Table 8 pharmaceutics-14-02170-t008:** Experimental table with independent variables and responses for the 2^4^ full factorial design with EC and E-RSPO.

Pattern	Independent Variables				Responses			
					EC		E-RSPO	
	PVA Conc. (% *w*/*v*)	Mixing Time (min)	Drug/ Polymer	Sonics (min)	Z-Ave (nm)	PDI	Z-Ave (nm)	PDI
**− + + −**	0.25	10	1:2	5	210.5	0.083	214.8	0.157
**+ − + −**	1	0	1:2	5	229.8	0.09	233.4	0.118
**− + + +**	0.25	10	1:2	15	363.6	0.245	320.2	0.334
**− − + +**	0.25	0	1:2	15	409.6	0.211	229.9	0.237
**− + − −**	0.25	10	1:1	5	353.5	0.218	403.3	0.635
**+ + − +**	1	10	1:1	15	193.5	0.072	241.6	0.168
**+ − − −**	1	0	1:1	5	208.4	0.108	243.2	0.141
**− − − +**	0.25	0	1:1	15	375	0.19	258.1	0.259
**+ − + +**	1	0	1:2	15	238.7	0.091	247.3	0.176
**+ + − −**	1	10	1:1	5	208.3	0.099	293.4	0.242
**− − − −**	0.25	0	1:1	5	460.9	0.201	230.3	0.221
**+ + + +**	1	10	1:2	15	235.6	0.118	249.1	0.178
**− − + −**	0.25	0	1:2	5	379.3	0.245	250.7	0.282
**− + − +**	0.25	10	1:1	15	365.2	0.216	245.1	0.266
**+ + + −**	1	10	1:2	5	229.8	0.117	242.5	0.133
**+ − − +**	1	0	1:1	15	198.5	0.096	237.3	0.18

**Table 9 pharmaceutics-14-02170-t009:** Secondary response surface design experimental table with the independent variables as well as responses obtained for EC and E-ESPO.

Formulation ID	Independent Variables			Responses—EC		Responses—E-RSPO	
	PVA conc. (% *w*/*v*)	Drug/ Polymer	EPV/ IPV	Z-Ave (nm) *	PDI	Z-Ave (nm)	PDI
**F-1**	0.5	2:1	10:4	NM	NM	1000	>0.5
**F-2**	1	1:1	10:2	214.3	0.197	118.3	0.083
**F-3**	0.25	3:1	10:2	450.9	0.423	169.8	0.253
**F-4**	1	1:1	10:4	289.6	0.371	132.7	0.085
**F-5**	0.25	3:1	10:4	NM	NM	187.3	0.147
**F-6**	1	1:1	10:4	178	0.168	130.9	0.123
**F-7**	0.25	1:1	10:2	365.6	0.395	152.8	0.088
**F-9**	0.5	2:1	10:4	171.2	0.149	1000	>0.5
**F-8**	1	3:1	10:6	NM	NM	144.3	0.115
**F-10**	0.25	1:1	10:6	NM	NM	286.2	0.277
**F-11**	1	3:1	10:2	142.7	0.106	682.1	0.291
**F-12**	1	3:1	10:2	145.2	0.071	116.7	0.138
**F-13**	0.5	2:1	10:4	191.5	0.209	140.5	0.108

* Z-ave responses that were too high to accurately measure were labelled “NM” (not measurable). PDI values for these trials could not be measured and were also labelled “NM”.

**Table 10 pharmaceutics-14-02170-t010:** The randomized order of experiments for the 20-run 3^2^ DoE with nanosphere CQAs responses.

Formulation		Independent Variables		Responses		
Pattern *	F	PVA Conc. (% *w*/*v*)	Drug/ Polymer	Z-Ave (nm)	PdI	Recovery (%)
**12**	F1	0.25	1:1	155.6	0.125	11.97
**21**	F2	1	1:4	145.9	0.083	34.52
**32**	F3	2	1:1	112.3	0.108	36.43
**23**	F4	1	4:1	101.4	0.069	36.25
**11**	F5	0.25	1:4	196.8	0.157	15.45
**22**	F6	1	1:1	127	0.11	26.92
**23**	F7	1	4:1	102.1	0.071	46.01
**22**	F8	1	1:1	125.6	0.08	27.06
**00**	F9	1.125	17:8	112.2	0.061	32.05
**32**	F10	2	1:1	111.2	0.121	37.78
**13**	F11	0.25	4:1	145.7	0.093	26.03
**21**	F12	1	1:4	148.1	0.124	33.37
**00**	F13	1.125	17/8	115.4	0.079	36.66
**33**	F14	2	4:1	109.3	0.254	39.6
**11**	F15	0.25	1:4	197.7	0.118	17.97
**12**	F16	0.25	1:1	165	0.182	16.4
**33**	F17	2	4:1	103.6	0.211	29.15
**31**	F18	2	1:4	110	0.071	60.22
**13**	F19	0.25	4:1	146.9	0.079	43.41
**31**	F20	2	1:4	107.3	0.054	54.65

* The pattern column represents low (1), mid (2), and high (3) levels of the parameters. The center points are represented as 00. F—formulation ID.

**Table 11 pharmaceutics-14-02170-t011:** Overall effect summary of the independent variables on the CQAs of the nanospheres.

Source	LogWorth	*p*-Value
PVA Conc. (0.25, 2)	5.248	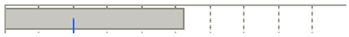	0.00001
PVA Conc.*Drug/Polymer	3.695	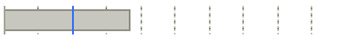	0.00020
Drug/Polymer (0.25, 4)	2.835	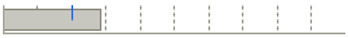	0.00146

The blue line shows the threshold based on the *p*-value.

**Table 12 pharmaceutics-14-02170-t012:** Comparison of CQA values predicted from the model against the validation formulation made with the optimum formulation parameters.

CQAs	Mean Predicted Values	Mean Optimized Formulation Values	*p*-Value
**Z-Ave (nm)**	105.8 ± 3.6	108.7 ± 1.7	0.2756
**PDI**	0.063 ± 0.013	0.063 ± 0.014	1.000
**% Recovery**	50.19 ± 8.00	57.43 ± 3.94	0.2324

## Data Availability

Data is contained within the article.

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
