# Peer review of "Quality by Design Guided Development of Polymeric Nanospheres of Terbinafine Hydrochloride for Topical Treatment of Onychomycosis Using a Nano-Gel Formulation"

_pharmaceutics, 2022, doi:10.3390/pharmaceutics14102170_

Round 1
Reviewer 1 Report
The study is successfully set up, but some experimental details are not clearly described or included.
1- You should include the LOQ and LOD determination in the analytical method, especially because you probably worked with low concentrations.
2- To talk about "entrapment efficiency", you should purify the nanoparticles from the not entrapped drug. If you don't remove this from the formulation, you calculate the total drug amount at the end of the preparation process. Still, you can't consider that all drug is inside the nanoparticles beforehand. This parameter is usually called "recovery". I suggest replacing entrapment efficiency with this or a similar word.
3- You sometimes mention the stability of the nanoparticles. Did you evaluate the physical stability, for example?
4- I didn't see any errors for the size, PdI, zeta-potential, or "entrapment efficiency" ..You should report each measurement with the corresponding standard error!
4- Check line 692
5- Check Figures 6 and 10 (too small and blurred)
Reviewer 2 Report
The abstract is well written and organized.
The study is interesting and well designed.
I suggest the following prior to acceptance:
Add “nanospheres” and “terbinafine” to the keywords
English editing is highly recommended to improve the style and punctuation, to reduce wordiness and in-direct meaning. It is difficult to read the current version.
Examples
· Please pay attention to the order of the terms in the sentence to make it easier to read, for example: “The most common type of skin disease worldwide are fungal infections of different types, especially 47 superficial infections – skin diseases being in the top 10 most prevalent [1,2]. One type of superficial 48 fungal infection, onychomycosis, is the most common disorder of the nail [3,4].” Moving “fungal infections” and “onychomycosis” to the beginning of the sentence will improve the style.
· Line 58 “Treatment of superficial fungal diseases has been challenging, due to factors such as uncertainty of 59 treatment duration, high relapse rates, and adverse effects of systemic treatments [5,11].” . “has been” should be “is”.
· Punctuation!!!!!!“It was hypothesized that the 109 positively charged nanospheres would get lodged into the structural deformities of hydrated 110 onychomycotic nails owing to their small size in addition to being bioadhesive on the negatively 111 charged nail plate favoring nail diffusivity [42,43].”
Line 79, add ethosomes as one of the most investigated carriers for topical delivery of antifungal agents [reference: https://doi.org/10.3390/molecules25132959].
Add statistics to figure 11
